# Estrogen-Related Receptor α: A Significant Regulator and Promising Target in Bone Homeostasis and Bone Metastasis

**DOI:** 10.3390/molecules27133976

**Published:** 2022-06-21

**Authors:** Chun Feng, Zhaowei Xu, Xiaojie Tang, Haifei Cao, Guilong Zhang, Jiangwei Tan

**Affiliations:** 1School of Pharmacy, Shandong Technology Innovation Center of Molecular Targeting and Intelligent Diagnosis and Treatment, Binzhou Medical University, Yantai 264003, China; fengchun0816@163.com (C.F.); zhaoweixv@bzmc.edu.cn (Z.X.); 2Department of Spinal Surgery, Yantai Affiliated Hospital of Binzhou Medical University, Yantai 264100, China; txjlfl@126.com (X.T.); chf1986@126.com (H.C.)

**Keywords:** ERRα, bone, osteoclast, osteoblast, bone metastasis

## Abstract

Bone homeostasis is maintained with the balance between bone formation and bone resorption, which is involved in the functional performance of osteoblast and osteoclast. Disruption of this equilibrium usually causes bone disorders including osteoporosis, osteoarthritis, and osteosclerosis. In addition, aberrant activity of bone also contributes to the bone metastasis that frequently occurs in the late stage of aggressive cancers. Orphan nuclear receptor estrogen-related receptor (ERRα) has been demonstrated to control the bone cell fate and the progression of tumor cells in bone through crosstalk with various molecules and signaling pathways. However, the defined function of this receptor in bone is inconsistent and controversial. Therefore, we summarized the latest research and conducted an overview to reveal the regulatory effect of ERRα on bone homeostasis and bone metastasis, this review may broaden the present understanding of the cellular and molecular model of ERRα and highlight its potential implication in clinical therapy.

## 1. Introduction

Bone is a hard and highly mineralized tissue, and it composes the major skeleton for body structure and provides a platform for essential physiological activity, such as exercise support and internal organ protection [1]. Moreover, the internal environment of bone is critical for the differentiation and maintenance of stem cells [2,3]. Bone homeostasis is the life-lasting balance between bone formation and resorption which involves two main cell types: hematopoietic original osteoclast, and mesenchymal stem cell oriented osteoblast [2]. Disruption between this equilibrium causes many functional disorder diseases, such as osteoarthritis, osteosclerosis, rhachitis, and osteoporosis [2,3]. Mechanism investigations have revealed that metalloproteinases, aging, estrogen signaling, dietary habit, and inflammatory cytokines may induce aberrant activation or repression of osteocyte and bone associated diseases [4,5,6]. Therefore, identifying more promising biomarkers for drug target will benefit the clinical treatment of osteopathia.

With the unique mineral content, high concentration of extracellular calcium, and specific microenvironment, bone is the most frequent area of lesions for metastasis in the progression of many carcinomas, especially hormone-sensitive breast cancer and prostate cancer [7,8]. Similar to metastasis events found in other organs, aggressive cancer cells firstly infiltrate local zones and complete the intravasation into circulation, then escape the immune surveillance, disseminate to appropriate ‘soil’, seed into bone marrow, then survive and become reactivated to form metastatic lesions in bone. The process of bone metastasis is regulated by key factors and signaling pathways related to metastasis and immune escape, and many hypotheses try to explain the mechanism of bone metastasis, such as cancer stem cell, epithelial-mesenchymal transition (EMT), circulating tumor cells (CTC), and cell plasticity remodeling. The molecular mechanism and pathological characteristics of bone metastasis have been systematically reviewed elsewhere [7,8,9], here we do not discuss these again.

Estrogen-related receptor α (ERRα) has been demonstrated to regulate the bone cell fate and bone metastasis in many types of cancers, but studies on its function and involved mechanism were inconsistent and controversial. Herein, a complementary literature search was performed in PubMed, EBSCO, Web of Science, and CNKI databases to obtain the relevant articles published before March 2022 without language restrictions using the relevant keywords. Moreover, the references cited by these articles were also read to identify additional relevant articles. Finally, a systematic review covering the recent findings was conducted to reveal perspectives on ERRα in bone homeostasis and bone metastasis.

## 2. Structure and Function of ERRα

ERRs belong to a subfamily of nuclear receptors and consist of three members: ERRα, ERRβ, and ERRγ. Owing to lack of endogenous ligands in vivo, they are also called orphan nuclear receptors [10]. ERRα was firstly identified from a cDNA library by using the DNA-binding domain (DBD) sequence of estrogen receptor α (ERα) as a probe [10]. Similar to other steroid hormone receptors, ERRα has an N-terminal located ligand-independent transcription activation domain, a centrally located DBD encompassing two zinc fingers that mediate the interaction between the specific DNA motif and the receptor, and a C-terminal located domain that is responsible for ligand dependent transaction (also called ligand binding domain (LBD)). Between the DBD and LBD, a poorly conserved hinge domain mainly mediates the homodimer formation [11,12], the schematic diagram of structure domains between human ERRs and ERα is illustrated in Figure 1. Despite the high structural conservation among ERRα and ERα, the ligand of ERα, such as 17β-estradiol, cannot activate its transcriptional activity. Subsequent research revealed that ERRα recognized and binded on the estrogen-related response element (ERRE) but not the estrogen r element (ERE) on the promoter region of target genes to activate the transcription expression [13]. However, many target genes of ERα (like *pS2* and *LTF*) were simultaneously regulated by the two receptors in breast cancer cells [14,15]. In addition, the transcriptional activity of ERRα is largely dependent on its co-regulators such as peroxisome proliferator-activated receptor γ coactivator-1 (PGC-1) α and PGC-1β, which is also distinguished from ERα that prefers to cooperate with SRC/p160 family [16].

ERRα has also been clarified to exhibit various regulatory functions in physiological and pathological processes, including energy metabolism [17], organ development, immune response [18,19], and tumorigenesis [20,21]. ERRα is strongly expressed in high energy consumption tissues (such as heart, liver, muscle, and adipose tissues), and it can regulate lipid metabolism and thermogenesis in both white adipose tissues (WAT) and brown adipose tissues (BAT) [22]. Mice with ERRα knockout (KO) rapidly exhibited hypothermia under low-temperature conditions, and the expression and activity of uncoupling protein 1 (UCP1) induced by adrenalin in BAT was largely dependent on the presence and activity of ERRα [23,24]. In addition, ERRα was proved to modulate glycolytic metabolism via regulating the expression of essential enzymes including phosphofructokinase, hexokinase 2, and enolase 1 [25]. ERRα cooperating with PGC-1α controlled the mitochondria metabolism and oxidative response to sensor the cellular ROS levels [26]. Beyond the function in metabolism, ERRα also acted in the innate immune response via regulating mitochondria ROS production and autophagy [18,19]. Furthermore, the high abundance of ERRα was frequently detected in many types of solid tumors (including colon, endometrium, bladder, ovary, breast, and prostate cancers), and its upregulation was associated with more aggressive phenotypes, drug resistance, and poor prognosis [20,27]. Mechanism exploration found that ERRα facilitated the proliferation and metastasis of tumor cells via upregulating oncogenes transcription and inducing metabolic reprogramming [21]. However, the role of ERRα in different cancers appears to be inconsistent. For example, ERRα almost functioned as an oncogenic factor in many types of carcinomas, but could serve as a favorable biomarker for the tamoxifen sensitivity in the patients with triple-negative breast cancer (TNBC) [28]. An alternative probability is that the functional performance of ERRα is tightly dependent on the specific cellular environment and post-translational modifications (PTMs) [27,29]. In addition, many specific inhibitors or natural products that function as agonists or antagonists of ERRα were identified, and these compounds have been implicated in the treatment of cancers or metabolism-disorder diseases [12,30]. Therefore, it is essential to explore the defined role of ERRα in a certain tumor model and achieve the precise therapy using the specific pharmacochemistry targeting on ERRα.

## 3. The Regulatory Role of ERRα in Osteoblasts

Epidemiological investigation showed that single nucleotide polymorphism (SNP) within *ESRRA* was closely associated with increased bone mineral density (BMD) of premenopausal women [31], and in situ hybridization assay found that *ESRRA* was consistently expressed in ossification zones during the mouse embryonic development stages, and it could directly bind to the promoter and activate the *OPN* transcription. These results indicated that ERRα may be involved in the process of the osteoblast progenitors differentiation and bone metabolism [32]. Indeed, ERRα was initially indicated to serve as a positive modulator for osteoblast differentiation and bone formation [33]; in vitro differentiation of osteoblast progenitors originating from rat calvaria or bone marrow was also accompanied with the continuous expression of ERRα, knockdown of ERRα by shRNA significantly impaired the proliferation and differentiation of osteoblast progenitors, while overexpression of this receptor increased the differentiation markers [34,35]. Moreover, estrogen could enhance the *ESRRA* transcription in a dose-dependent manner in the early differentiation stage of osteoblast progenitor cells [36,37], inhibition of ERRα in a culture medium induced a decreased colony number of mineralized cells [34], and the progenitor cells with ERRα deficiency were inclined to differentiate into adipocytes as evidenced through the increasing lipid droplet accumulation and expression of adipocyte markers (such as PPARγ or aP2). These studies suggested that ERRα served as a switch factor to favor the precursor cells differentiating into osteoblasts but not adipocytes [37,38]. Furthermore, other models from mice and humans also supported the promoted effect of ERRα on the osteoblast differentiation and functional exhibition, Rajalin et al., isolated the mouse mesenchymal stem cell from bone marrow and conducted ex vivo differentiation culture; they found that knockdown of ERRα induced the decreased cell proliferation, osteoblastic differentiation, and mineralization with reduced expression of bone sialoprotein (BSP) [39]. Cai et al., demonstrated that the mRNA and protein levels of ERRα were significantly increased during the late stage of osteogenic differentiation of human periodontal ligament cells (hPDLSCs), and the receptor maintained the osteogenic differentiation via regulating the ALP activity and osteogenesis-related genes expression [40], and another group established that ERRα interacted cooperatively with PGC-1α in the promoter region of *osteocalcin* and enhanced its transcription expression to facilitate the osteogenesis [41]. In the human MSCs model, Huang et al., demonstrated that ERRα could directly bind to ERRE on the *Gls* promoter and transactivate its transcription, then further regulated glutamine anaplerosis and mitochondrial function, and this effect was required for osteogenic differentiation and impaired by aging [42]. Overall, these data indicated that ERRα functioned as a driving factor for osteoblast differentiation and was resistant to bone loss.

However, an increasing number of studies proposed that ERRα preferred to repress osteoblast differentiation and caused bone loss in vitro and in vivo. Firstly, the association between *ESRRA* SNP and increased BMD was not replicable in another set of independent clinical data [43]. Two different groups consistently demonstrated that ERRα-KO mice exhibited increased BMD and enhanced osteoblast activity when compared with control, osteoblast progenitors derived from mice with ERRα deficiency had elevated expression of differential markers and were more liable to differentiate to matured phenotype but not adipocyte lineage [44,45]. One of groups further showed that the animals with ERRα-KO had resistance to the bone loss induced by aging or estrogen withdrawal mimicked with ovariectomy, indicating that targeting the receptor in menopause could prevent excessive bone loss and reduce the rate of fracture [45]. Moreover, Gallet et al., generated a conditional *ESRRA*-KO mouse in which ERRα was normally presented during the early differentiation stage of osteoblasts but lost its function in maturation; they found that ERRα could effectively respond to estrogen deficiency and inhibit the osteoblast differentiate and osteogenesis in the later maturation stage but not the early stage [46]. This phenomenon was consistent with the regulatory role of ERRα investigated in the mineralization of osteoblasts [44,47]. OPN had been verified to be the inhibitor of mineralization and one of the target genes of ERRα, ERRα deficiency was accompanied with a dramatic reduction of OPN and impairment of mineralization [48]. Consistently, *OPN*-/- mice were also resistant to the bone loss induced by estrogen withdrawal, this phenotype was similar to ERRα-KO animals [49]. Mechanism investigation revealed that ERRα might serve as a dual functional factor to modulate the activation of signaling pathways and osteogenesis-related genes expression (such as *Runx2* and *OPN*), and this effect was mainly dependent on the presence of its cofactor PGC-1β [50,51]. Studies using different models, such as different cell lines or mouse strains, may obtain inconsistent and contrary results (Figure 2). Therefore, more comprehensive research deserves to be conducted to uncover the defined role of ERRα in osteoblast differentiation and bone homeostasis.

## 4. The Role of ERRα in Osteoclasts

Bone remodeling is not only associated with osteoblasts-mediated bone formation, but involves bone resorption regulated by the osteoclasts. Bonnelye et al., conducted the in vitro culture experiments and revealed that ERRα exhibited little effect on osteoclast differentiation but promoted adhesion and migration of osteoclasts via regulating the expression of OPN and integrin β3 chain and the alteration of cytoskeleton [52]. This conclusion was partially consistent with the role of ERRα in breast cancer [20]. However, in vivo studies using ERRα-KO animals challenged this conclusion. They found a specific phenotype with increased bone formation and reduced bone absorption in an *ESRRA*-/- mouse model owing to the decreased number and activity of osteoclasts [44,53]; mechanism investigation revealed that ERRα cooperated with PGC-1β and controlled the mitochondria biosynthesis and oxidative activity, which indirectly enhanced the osteoclast differentiation [53]. They further demonstrated that ERRα served as the key factor in modulating the RANKL- and rosiglitazone-mediated bone loss and functional performance of osteoclasts [53]. Consistently, PGC-1β was parallel with ERRα expression during the osteoclasts transition and the knockout of PGC-1β in mice exhibited osteopetrosis with abnormal morphology and the impaired bone-resorption activity of osteoclasts [54]. Subsequently, the significant role of ERRα in cholesterol metabolism and bone homeostasis was confirmed by Wei et al. [55] who identified that cholesterol acted as an endogenous ligand for ERRα to activate its transcriptional activity; this effect induced the inhibition of proinflammatory cytokinesis production by macrophage and facilitated the osteoclast differentiation [55]. Concomitantly, pharmacological inhibition on cholesterol synthesis and bone resorption by statin and bisphosphonate were also modulated by ERRα [55]. Considering the origin of osteoclast and macrophage from embryogenesis, ERRα may act as rheostat factor in the differentiation of macrophage and osteoclast. Congruously, MYC promoted the oxidative metabolism and drove osteoclastogenesis, the process of which was largely dependent on the ERRα action [56]. These fundamental studies suggested that the inhibition of ERRα by molecules or natural products may prevent or improve the bone loss induced by aging or hormone withdrawal. Indeed, Zheng et al., identified that carnosic acid, isolated from *Rosmarinus officinalis* L., could decrease the transcription activity of ERRα via reducing the interaction between ERRα and PGC-1β and triggering proteasomal degradation of ERRα, then suppress RANKL-mediated osteoclastogenesis and ovariectomy-induced bone loss [57]. Andrographolide, another natural compound targeted on ERRα from virtual docking screening, had been proven to suppress the formation of ERRα/PGC-1β complex and further reduce glutaminase metabolism during osteoclast differentiation. Inhibition on this regulatory axis attenuated osteoclasts-mediated bone resorption and bone loss in vivo [30]. Taken together, these data illustrated that ERRα functioned as a promoting effector to facilitate osteoclast differentiation and bone absorption involving various mechanism networks (Figure 2), targeting on the receptor would be a promising strategy for degenerative bone disorders.

## 5. The Role of ERRα in Cartilage and Osteoarthritis

Osteoarthritis (OA) is characteristically associated with chronic inflammatory pathological feature and closely associated with articular cartilage dysfunction (including abnormal proliferation and erasion of cartilage) [38,58,59]. During the development of osteoarthritis, the elevated iNOS, IFNs, and TNFs levels in the microenvironment were detected, and LPS or TNF-γ induced the aberrant activation of NF-κB signaling in the pro-inflammatory (M1) macrophages, which could generate pro-inflammatory cytokinesis (such as TNF-γ, IL-6 and IL-12) to form a positive feedback loop and result in the degradation of cartilage [60]. ERRα has been reported to be involved in the innate immune response and functional performance of macrophage via modulating autophagy or TLR4/TRAF6/NF-κB axis [19,61]. Consistently, ERRα also performed a dual role in cartilage formation and OA progression. The effect of ERRα on cartilage formation was initially involved in the regulation of SOX-9, Bonnely et al., reported that the receptor upregulated Sox-9 transcription and induced the differentiation and proliferation of chondrocytes [62]. The chondrocytes isolated from the OA patients treated with XCT790 (the specific reverse agonist of ERRα) showed the downregulation of SOX-9 expression in a dose dependent manner. These results were proven by Kim et al., in a zebrafish embryo model, where ERRα knockdown led to the malformation of pharyngeal arch cartilage [63]. In addition, ERRα also contributed to the cartilage degradation which is associated with IL-1β and MMP-13. IL-1β stimulation could increase the ERRα expression via the PGE2/cAMP/PKA signaling pathway, ERRα reversely activated the MMP-13 expression in OA progression [64]. This process may involve the macrophage action mediated by ERRα/NF-κB signaling, but this possibility needs to be further investigated. The comprehensive effect of ERRα on cartilage and OA had been reviewed by Bonnelye and Tang [38,65], but more systematic research is indispensable to reveal the crosstalk between inflammatory signaling pathways and ERRα action in cartilage.

## 6. The Role of ERRα in Bone Metastasis

According to the different mechanisms involved, bone metastasis is mainly classified into two types: osteoblastic (bone-forming) metastasis and osteolytic (bone-resorbing) metastasis, which are closely associated with abnormal activity of osteoblasts or osteoclasts, respectively. Considering the essential role of ERRα in cancer metastasis and osteocyte differentiation (Figure 2), the role of ERRα in bone metastasis was also investigated. Similar to the complicated action of ERRα in bone metabolism, its role in bone metastasis was also controversial. Although the studies regarding the regulation of ERRα in bone metastasis were mainly produced from the same research group, the conclusion appears to be evolutionary and inconsistent (Figure 3). Initially, ERRα was shown to enhance the growth in situ but repress the proliferation of breast cancer cells in bone via upregulating the osteoclastogenesis inhibitor OPG and angiogenic growth factor VEGF in a mouse xenograft model of metastatic human breast cancer. A primary breast tumor with overexpression of ERRα exhibited highly vascularized and increased growth ability, while the receptor inhibited the differentiation and activity of osteoclasts, and further limited the tumorigenesis in bone [66]. However, the study conducted in castration-resistant prostate cancer (CRPC), which had higher aggression and frequently occurring bone metastasis, obtained a distinctive conclusion: they found that the overexpression of ERRα significantly increased the progression of CRPC and metastasis in bone, and mechanism investigation revealed that the receptor enhanced the bone remodeling via increasing the expression of metastatic factors (such as VEGF-A, WNT5A, TGFβ1, and periostin) and generating a favoring-growth tumor environment [67]. In addition, Vargas et al., also found that ERRα promoted orthotopic breast cancer cell dissemination to bone but not lung metastasis in a mouse breast cancer model through increasing the expression of RANK, which acted as the receptor of RANKL and guided the cancer cells’ migration into the bone microenvironment. In line with this, RANKL stimulation induced a higher activity of mTOR signaling in 4T1 cells with ERRα overexpression, and pharmaceutical inhibition of ERRα by C29 complex inhibited the tumorigenesis and bone metastatic events [68]. Interestingly, Bouchet et al., subsequently revealed that ERRα inhibited the growth of breast cancer cells after tumor cell anchorage in the bone via activating the immune response in bone microenvironment. This phenomenon could be achieved via ERRα-mediated induction of chemokines (CCL17 and CCL20) and reduction of TGFβ3, which recruited the CD8 positive T cells to trigger immune cytotoxic action and metastasis repression [69]. The reason that ERRα appears to function inconsistently or reversely in different cancer types or stages of bone metastasis is a critical issue to be explored. One alternative probability is that prostate cancer predominantly prefers osteoblastic lesions, and the metastasis from breast cancer mostly exhibits osteolytic lesions [70], which may explain the diverse role of ERRα between prostate cancer and breast cancer. With regard to the different function of ERRα between dissemination of tumor cells to bone and the growth of metastases, the distinguished microenvironment and signaling stimulation may account for these results [8,9]. Therefore, an appropriate metastatic model and comprehensive analysis of regulatory network will illustrate the important effect of ERRα on the initiation and progression of bone metastasis.

## 7. Conclusions

This review discussed the regulatory role of ERRα in bone metabolism and bone metastasis, from which we could conclude that ERRα exhibited diverse functions and regulated multiple pathways associated with bone fate and metastasis, such as differentiation of osteoblast and osteoclast, macrophage-mediated immune response, and hormone-mediated metabolism reprogramming. These studies not only indicated the promising potential of ERRα as the drug target in clinical treatment of bone disorder diseases, but also reminded us that the single suppression of ERRα by specific inhibitors or natural compounds may induce potential side-effects. Therefore, more comprehensive investigation regrading ERRα in a certain type of bone-related diseases or stage of progression should be conducted to achieve precise treatment.

## Figures and Tables

**Figure 1 molecules-27-03976-f001:**
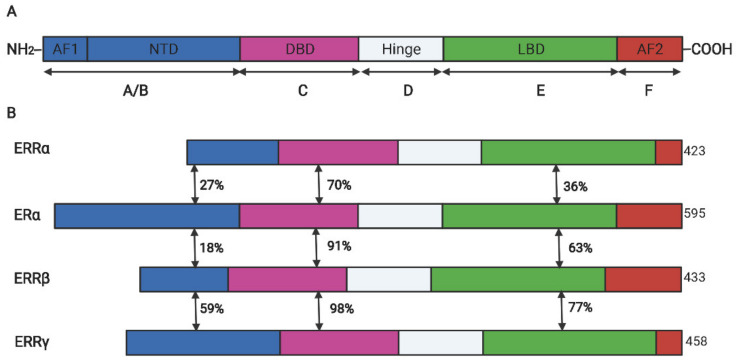
The schematic diagram for the protein domains of ERα and ERR isoforms. (**A**). The conserved structure of nuclear receptors. (**B**). The domain similarity between ERRs and ERα. DBD, DNA binding domain; LBD, ligand binding domain; AF, activation functional domain; NTD, N-terminal domain; The double-headed arrows indicated the sequence similarity between two receptors.

**Figure 2 molecules-27-03976-f002:**
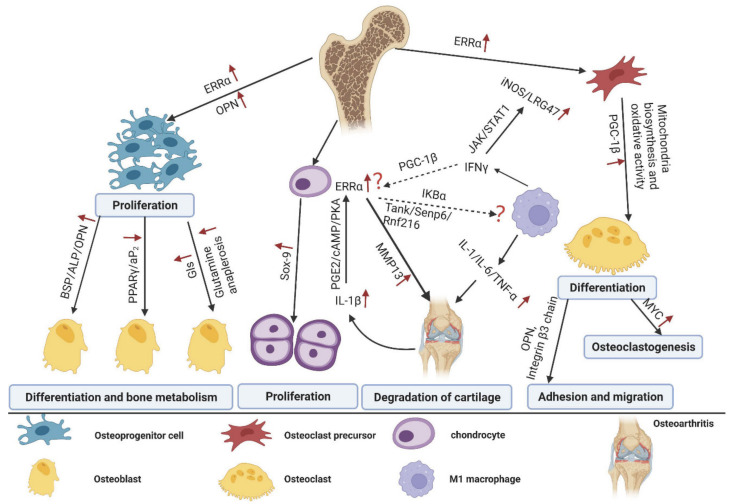
The regulatory mechanism of ERRα in the function of bone associated cells and bone metabolism. On the one hand, ERRα impacts osteoblast development and metabolism by upregulating BSP/ALP/OPN, Gls-mediated glutamine anaplerosis and inhibiting PPARγ/aP2, which ultimately results in the differentiation and proliferation of osteoblasts. On the other hand, ERRα interacts with PGC-1β to promote osteoclast formation, adhesion, and migration through regulating the expression of OPN and integrin β3, mitochondrial metabolism and oxidative processes in a synergistic manner. In the cartilage, upregulation of SOX-9 induced by ERRα promotes chondrocyte proliferation, while ERRα expression could be activated through the PGE2/cAMP/PKA signaling pathway in response to IL-1β stimulation, the receptor modulated the synthesis of MMP-13 and caused osteoarthritis. Elevated iNOS or cytokines including IFN-γ and TNF-γ could activate M1 macrophages to release IL-1β, this positive feedback loop might lead to cartilage erosion and osteoarthritis. However, the interactive modulation between ERRα and macrophages in the cartilages is still unclear.

**Figure 3 molecules-27-03976-f003:**
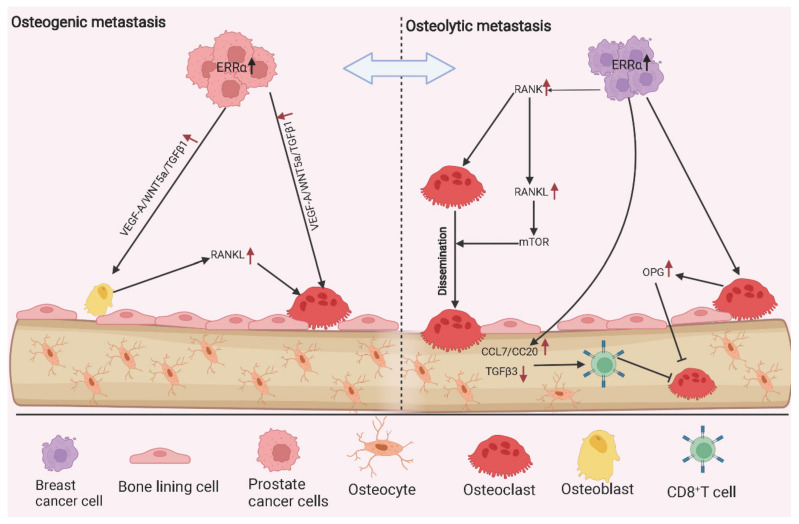
The functional role of ERRα in osteogenic and osteolytic metastases. Both osteogenic and osteolytic metastases are involved in the interactions among tumor cells, osteoblasts and osteoclasts, osteogenic and osteolytic metastases can reciprocally transit. In the osteogenic lesions, ERRα overexpression in prostate cancer cells directly or indirectly modulate bone re-modeling by boosting the expression of metastasis factors (VEGF-A, TGFβ1, WNT5A). The interactive modulation between osteoblasts and osteoclasts generates a tumor-friendly environment to facilitate tumor anchoring and growth. In the breast cancer, ERRα promotes the cell growth in situ but inhibits the proliferation of breast cancer cells in bone via increasing angiogenesis and exacerbating bone environment (OPN-mediated growth repression and cytokines-induced immune activation). Moreover, ERRα also facilitates the breast cancer dissemination to the bone through upregulating RANK and activating mTOR signaling pathway.

## Data Availability

Not applicable.

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
