# Peer review of "Estrogen-Related Receptor α: A Significant Regulator and Promising Target in Bone Homeostasis and Bone Metastasis"

_molecules, 2022, doi:10.3390/molecules27133976_

Round 1

Reviewer 1 Report

The paper is a review of the role of Estrogen-related receptor alpha (ERRa) in bone cells and how the receptor interacts with these bone cells and seems to influence bone disease and metastasis.  Because of the conflicting results published in the scientific literature and the many purported roles of ERRa, the area is an important one.  The authors present interesting and relevant information, but the paper can be improved.  

1. The figure legends should provide more information describing the specific figure.  The figure diagrams are very interesting and helpful, but need a more detailed description with them.

2. There are several typos and incorrect spellings throughout the paper that should be corrected (e.g. rhachitis should be rachitis and RNAKL should be RANKL  etc...).

3. It also would be helpful to define terms.  For example, the word arthrophlogosis (in Intro section) could not be found online or in Taber's Medical Dictionary.

4. There are several places where the grammar or sentence structure could be improved.  Several sentences are either awkward or not clear.

With some extensive editing, the paper has great potential.

Author Response

  1. The figure legends should provide more information describing the specific figure.  The figure diagrams are very interesting and helpful, but need a more detailed description with them.

Response: Thanks for the reviewer’s recognition for our work. According to the reviewer’s suggestion, we have detailed the description of all the three figure diagrams and summarized the key explanation. Moreover, we also redrew the Figures to provide more precise information. Please check them in our revised manuscript.

  1. There are several types and incorrect spellings throughout the paper that should be corrected (e.g. rhachitis should be rachitis and RNAKL should be RANKL  etc...).

Response: Thanks for the reviewer’s careful reading for our manuscript. We have carefully checked our manuscript and corrected the spelling errors, and the revised sections were marked with “track changes” in our new version of manuscript.

  1. It also would be helpful to define terms.  For example, the word arthrophlogosis (in Intro section) could not be found online or in Taber's Medical Dictionary.

Response: Thanks for the reviewer’s suggestion, we have replaced “arthrophlogosis” with “osteoarthritis”, we also checked the terms used in our manuscript and ensured their scientificity and universality.

  1. There are several places where the grammar or sentence structure could be improved.  Several sentences are either awkward or not clear.

Response: As the reviewer suggested, we have carefully checked and revised the grammar errors or inappropriate sentence structure, and polished the language and expression to improve our manuscript.

Reviewer 2 Report

This review paper focuses on the role and mechanisms of ERR-alpha in bone metabolism and cancer bone metastasis. The contents are updated, well-organized, and balanced. However, there are many spelling and grammatical errors. A careful proofreading and editing are needed to improve the quality and scientific accuracy of the paper. 

Author Response

Reviewer #2

This review paper focuses on the role and mechanisms of ERR-alpha in bone metabolism and cancer bone metastasis. The contents are updated, well-organized, and balanced. However, there are many spelling and grammatical errors. A careful proofreading and editing are needed to improve the quality and scientific accuracy of the paper.

Response: Thanks for the reviewer’s recognition for our paper, we have carefully proofread and edited the manuscript to improve our paper. The incorrect spellings and grammatical errors were all corrected and they were marked up in our revised manuscript.

Reviewer 3 Report

The review is comprehensive and relevant. The review article includes a clear and concise abstract. The introduction sets the scene by describing all the recent finding to uncover the perspective of ERRα in the bone homeostasis and bone metastasis. The authors elucidated the regulatory role of ERRα in bone metabolism and bone metastasis, from which they conclude that ERRα exhibited diverse function and regulated multiple pathways. This key message is conveyed through the text, conclusion and figures. 

General comments:

1. In the subheading The role of ERRα in cartilage and osteoarthritis it should address the other inflammatory markers (iNOS, NFkB, TNFa) that might be involved in OA and also to give overview with relation to macrophages involvement.

2. The methodology for literature search is not described.

 Minor comments:

1. The spell check and grammar should be performed. For example, in introduction space is missing before reference in bracket or space is missing after interpunction sign”.”.

Author Response

  1. In the subheading “The role of ERRα in cartilage and osteoarthritis” it should address the other inflammatory markers (iNOS, NFkB, TNFa) that might be involved in OA and also to give overview with relation to macrophages involvement.

Response: As the reviewer suggested, we have read the studies about the inflammatory markers that might be involved in OA (such as iNOS, TNFs and IFNs) and macrophage polarization. And we also discussed this issue in the section subheading “The role of ERRα in cartilage and osteoarthritis”. Moreover, we cited these researches as Ref #58, #59, #60 and #61 in our revised manuscript. Please check it at Page 6.

  1. The methodology for literature search is not described.

Response: Thanks for the reviewer’s suggestion, we have added the methodology for literature search in the Introduction section: “Herein, a complementary literature search was performed in PubMed, EBSCO, Web of Science and CNKI databases to obtain the relevant articles published before March 2022 without language restrictions using the relevant keywords. Moreover, the reference cited by these articles were also read to get additional relevant articles. Finally, a systematic review covering the recent findings was conducted to reveal the perspective of ERRα in the bone homeostasis and bone metastasis”. Please check it in our revised manuscript at Page #2 line 6-11.

 Minor comments:

  1. The spell check and grammar should be performed. For example, in introduction space is missing before reference in bracket or space is missing after interpunction sign”.

Response: Thanks for the reviewer’s careful reading for our paper, we have checked the errors and revised them in our manuscript to improve the paper quality.

Round 2

Reviewer 1 Report

Addressed concerns adequately.